# Morphology, Taxonomy, Anatomy, and Palynology of the Opium Poppy (*Papaver somniferum* L.) Cultivation in Northern Thailand

**DOI:** 10.3390/plants12112105

**Published:** 2023-05-25

**Authors:** Chatchai Ngernsaengsaruay, Nisa Leksungnoen, Pichet Chanton, Tushar Andriyas, Pratchaya Thaweekun, Surasak Rueansri, Ratthapon Tuntianupong, Woottichai Hauyluek

**Affiliations:** 1Department of Botany, Faculty of Science, Kasetsart University, Bangkok 10900, Thailand; pichet.cha@ku.th; 2Biodiversity Center, Kasetsart University (BDCKU), Bangkok 10900, Thailand; 3Department of Forest Biology, Faculty of Forestry, Kasetsart University, Bangkok 10900, Thailand; thugnimics28@gmail.com; 4Narcotics Crop Survey and Control Institute, Office of the Narcotics Control Board (ONCB), Ministry of Justice, Chiang Mai 50300, Thailand; pratchaya.fus@gmail.com (P.T.); surasakr54@gmail.com (S.R.); matekoog.skk@gmail.com (R.T.); woottichai.hl@gmail.com (W.H.)

**Keywords:** opium alkaloids, Papaveraceae, Papaveroideae, pollen morphology, poppy seeds, Ranunculales

## Abstract

In this paper, we present the morphology, taxonomy, anatomy, and palynology of *Papaver somniferum*. A detailed morphological description and illustrations of the species are provided, along with information about the identification, distribution, cultivation areas, habitats, pollinators, specimens examined, growing periods, phenology, etymology, vernacular name, and uses. The species can be characterized as a glabrous and glaucous herb with unlobed or pinnately lobed leaves, and an amplexicaul base; variations in color and morphological characteristics of petals; and white filaments, occasionally purple with a white basal part, broadened at the apical part. Two rings of discontinuous and widely spaced collateral vascular bundles are recognized in the transverse section of the stems. The shape of epidermal cells on the adaxial surface is polygonal, while that on the abaxial surface is polygonal or irregular. The anticlinal cell walls of the epidermal cells on the adaxial surface are straight or slightly curved, while those on the abaxial surface are straight, slightly curved, sinuate, or strongly sinuate. The stomata are anomocytic and are confined to the lower epidermis. The stomatal density is 54–199/mm^2^ (with a mean of 89.29 ± 24.97). The mesophyll is not distinctly differentiated into palisade and spongy regions. Laticifers are found in the phloem areas of the stems and leaves. The pollen grains can be spheroidal or prolate spheroidal in shape, sometimes oblate spheroidal [polar axis (P) diameter/equatorial axis (E) diameter ratio = 0.99–1.12 (with a mean of 1.03 ± 0.03)]. The pollen aperture is tricolpate and the exine sculpturing is microechinate.

## 1. Introduction

*Papaver* L. belongs to a group of annual and perennial herbs in the subfamily Papaveroideae Eaton under the family Papaveraceae Juss., and the order Ranunculales Juss. ex Bercht. & J. Presl [1,2]. The genus consists of 149 accepted species and is distributed in the temperate and subtropical regions of the Northern Hemisphere and up to South Africa [3]. The genus is characterized as a group of herbaceous annual and perennial plants producing latex; they usually have sessile leaves, which can sometimes have an amplexicaul base; drooping flower buds; caducous sepals and petals; and petals in two whorls. The outer whorl is larger. It usually has numerous stamens; a one-locule ovary, with numerous ovules; and radiate and sessile, actinomorphic stigmas, united into a compressed, lobed disk. The fruit is usually a poricidal capsule that has numerous, small, reniform seeds, which are longitudinally striated, alveolated, or pitted [4,5,6,7,8,9,10,11].

Opium poppy is reported to have originated in ancient Mesopotamia (modern day Iraq and Kuwait) [12] and been sourced for latex (milky sap); its alkaloids are used in the pharmaceutical industry or for poppy seeds which are used in the food industry [13]. Opium poppy is illegal in most parts of the world, including Thailand. There are only a few countries where the cultivation of opium poppy is legal for the pharmaceutical and food industry, such as in Australia, Canada, India, Central and Southern America, Türkiye (Turkey), Russia, the Czech Republic, Slovakia, Holland, France, Hungary, Iran, Poland, Romania, and Spain [14,15].

Different cultivars with low alkaloid content can be used in the food industry for seed or oil production [16], such as in some European countries (e.g., Germany and the Czech Republic), which only permit the cultivation of ‘low-morphine’ varieties of *Papaver somniferum*. Apart from these, some countries only grow varieties to derive pharmaceutical use from the capsules (e.g., Australia, France, and Spain) or for both culinary and pharmaceutical uses (e.g., Hungary and Slovakia). Opium poppy grows on a wide variety of soil textures, but the clayey-type soils can be relatively harder to plough and pulverize sufficiently for the roots of young opium poppy plants to penetrate. On the other hand, sandy soils tend to lose water to percolation, resulting in the moisture being insufficient for healthy growth.

Opium poppy prefers moderately cool weather and an open sunny location; severe cold spells, frost, dull cloudy weather, high winds, and very heavy rainfall during the lancing period adversely affect the quantity and quality of opium yield [17]. Water stress can affect the alkaloid production during various developmental stages of opium poppy plants, with sufficient supply water being beneficial for alkaloid accumulation in the capsules, while drought can increase the level of certain alkaloids. Nitrogen fertilization can elevate alkaloid accumulation [18] only under excessive light conditions, while severe drought can reduce the accumulation of morphinans [19].

Under pressure of the Nixon administration′s “war on drugs”, a crackdown on opium poppy cultivation was undertaken by the Thai government, deeming it illegal to grow the crop, with a reported drop in production from well over 10,000 ha in 1961 to under 300 ha in 2015 [20]. Subsequent research and development of geographically suitable alternative crops and other incentives meant that small landholders were dissuaded from cultivating the crop. This illegal status of the opium poppy crop meant that no substantial research could be conducted about the plant and its medical benefits in Thailand. Additionally, there are almost no data about the morphological traits of opium poppy in Thailand and the herbarium specimens are already outdated. With proven benefits of opium poppy already reported, the Office of the Narcotics Control Board, Thailand, plans to study opium poppy for further usage in the pharmaceutical industry. In light of that effort, this study is the first comprehensive study of its kind reporting on the morphological characteristics of opium poppy growing naturally in Thailand. Thus, this study will add to the fundamental knowledge about the species in Thailand for future uses in the medical industry.

*Papaver somniferum* L. (opium poppy) has long been illegally cultivated in the Golden Triangle, or the trijunction of Myanmar, Lao PDR, and Thailand along the Mekong River, and has long been associated with illegal opium production and drug trafficking. Opium poppy in South East Asia is mostly cultivated on steep mountainous terrain [21]. The illegal cultivation of opium poppy remains a serious problem in Thailand, even though it is mostly found along the slopes of the hilly areas of nine northern provinces and one northeastern province. The main areas of cultivation are located in the Chiang Mai, Tak, and Mae Hong Son provinces (from Narcotics Crop Survey and Control Institute, Office of the Narcotics Control Board [ONCB] observations, 2020–2023).

Several studies related to *Papaver somniferum* have been reported previously from other countries [2,4,5,6,7,8,9,10,15,21,22,23,24,25,26,27,28]. However, due to its illegal status in Thailand, such information about opium poppy is mostly lacking, and its characteristics are poorly known. Therefore, in this paper, we provide knowledge relevant to opium poppy grown in Thailand, which was obtained from the research project entitled “Morphology, Taxonomy, Anatomy, and Palynology of the Opium Poppy (*Papaver somniferum* L.), Papaveraceae, Cultivation in Northern Thailand”. We have obtained permission to study and support from the Narcotics Crop Survey and Control Institute, ONCB, Ministry of Justice.

## 2. Results

### 2.1. Morphology and Taxonomy

#### *Papaver somniferum* L., sp. Pl. 1: 508. 1753. (Figure 1, Figure 2, Figure 3, Figure 4, Figure 5, Figure 6, Figure 7, Figure 8 and Figure 9)

*Papaver somniferum* L., sp. Pl. 1: 508. 1753; Hook. f. and Thomson in Hook. f., Fl. Brit. India 1: 117. 1872; Craib, Fl. Siam. 1(1): 75. 1925; Backer and Bakh. f., Fl. Java (Spermatoph.) 1: 178. 1963; D. G. Long in Grierson and D. G. Long, Fl. Bhutan 1(2): 401. fig. 32h. 1984; Kiger and D. F. Murray, Fl. N. Amer. 3: @eFloras.org. 2008; M. Zhang and Grey-Wilson in C. Y. Wu, P. H. Raven and D. Y. Hong, Fl. China 7: 278. 2008; P. A. Egan, Pendry and S. Shrestha in M. F. Watson et al. Fl. Nepal Webedition 1: 3. 2012; Suddee in Chayam. and Balslev, Fl. Thailand 14(3): 505. 2019. Type: South Europe (lectotype LINN [Herb. Linn. no. 669/8], designated by S. M. H. Jafri and M. Qaiser, Fl. W. Pakistan 61: 20. 1974, not seen.

**Figure 1 plants-12-02105-f001:**
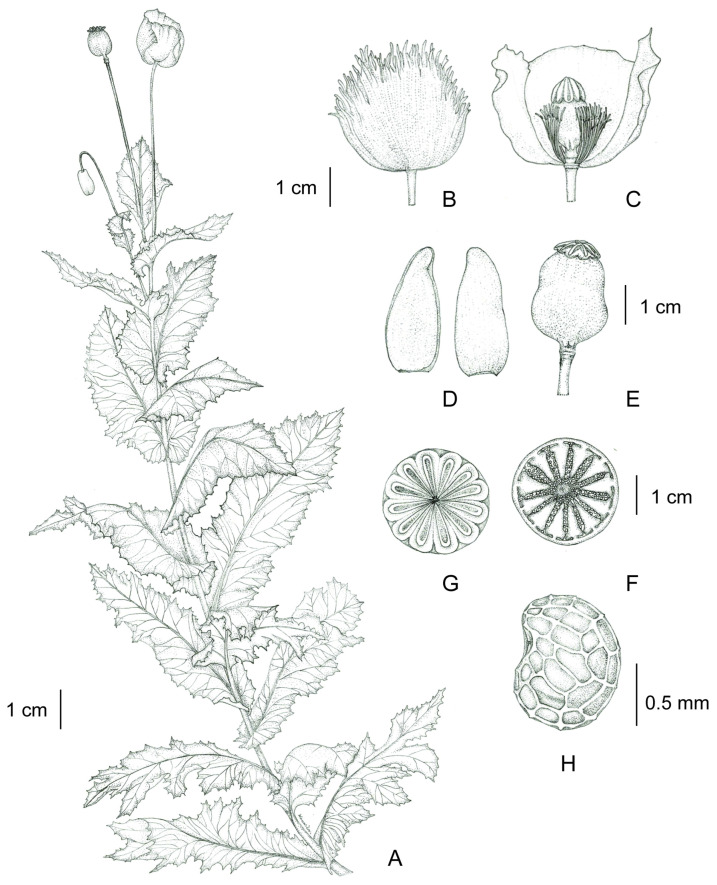
*Papaver somniferum*. (**A**) Fertile branch; (**B**) flower; (**C**) opened flower showing stamens and pistil (side view, an outer petal removed); (**D**) sepals: inside (left) and outside (right); (**E**) fruit with radiating stigmatic rays, united into an enlarged, persistent, sessile stigmatic disk and a constricted stipitate; (**F**) transverse section of fruit; (**G**) an enlarged, persistent, sessile stigmatic disk with radiating stigmatic rays (top view); and (**H**) seed showing faveolated and a minutely pitted surface (as observed under a stereo microscope). Materials from *C. Ngernsaengsaruay* et al. *Ps03-06012023* (BKF) and *C. Ngernsaengsaruay* et al. *Ps04-06012023* (BKF). Drawn by Wanwisa Bhuchaisri.

**Figure 2 plants-12-02105-f002:**
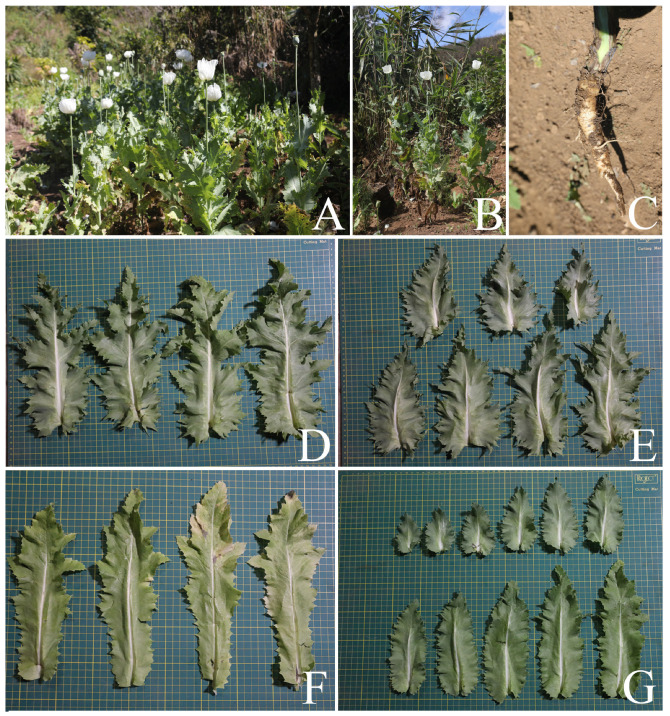
*Papaver somniferum*. (**A**,**B**) Habit; (**C**) taproot; (**D**–**F**) leaf shapes: lanceolate, (**G**) lanceolate-ovate, ovate, and broadly ovate. Photos: Chatchai Ngernsaengsaruay.

**Figure 3 plants-12-02105-f003:**
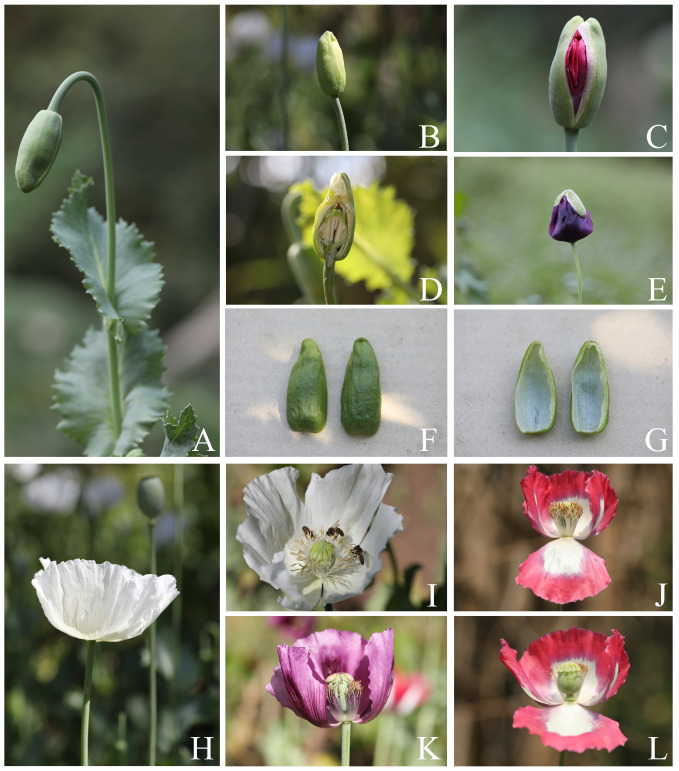
Reproductive parts of *Papaver somniferum*. (**A**) Drooping flower bud; (**B**) erect flower bud before anthesis; (**C**) nearly opened flower bud; (**D**) longitudinal section of flower bud; (**E**) nearly opened flower and sepals separated from the base to the apex (sepals falling off before being fully opened flower); (**F**) sepals (outside); (**G**) sepals (inside); (**H**) opened flower; (**I**) flower showing stamens and pistil (top view); (**J**) flower showing stamens with white filaments and longer than the pistil (side view); (**K**) flower showing stamens, purple with a white basal part of filaments, and pistil (side view, some stamens and an outer petal removed); and (**L**) flower showing stamens and pistil (side view, some stamens and an outer petal removed). Photos: Chatchai Ngernsaengsaruay.

**Figure 4 plants-12-02105-f004:**
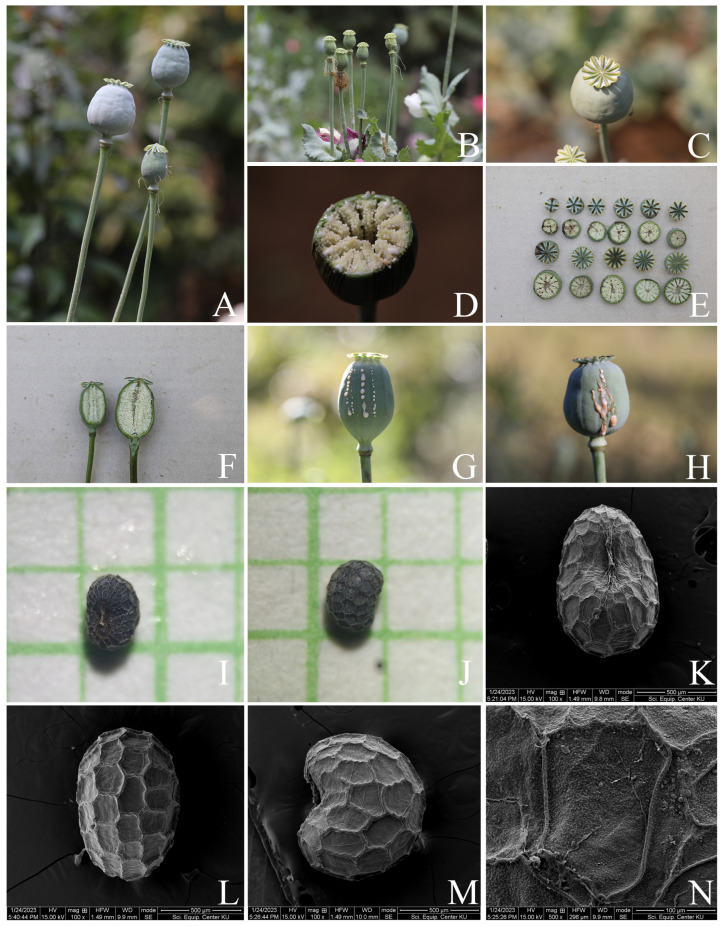
Reproductive parts of *Papaver somniferum*. (**A**) Pistil (petals and stamens falling off) and fruits; (**B**) pistils (petals and stamens falling off); (**C**) fruit (top view); (**D**) transverse section of fruit; (**E**) transverse section of fruits showing carpels, ovules, parietal placentation, and radiating stigmatic rays, united into an enlarged, persistent, sessile stigmatic disk; (**F**) longitudinal section of fruit; (**G**,**H**) latex secreted from cut fruits; (**I**,**J**) LM micrographs of seeds; and (**K**–**N**) SEM micrographs of seeds showing faveolated and a minutely pitted surface. Photos: Chatchai Ngernsaengsaruay (**A**–**H**); Pichet Chanton (**I**,**J**); and Scientific Equipment Centre, Faculty of Science, Kasetsart University (**K**–**N**).

**Figure 5 plants-12-02105-f005:**
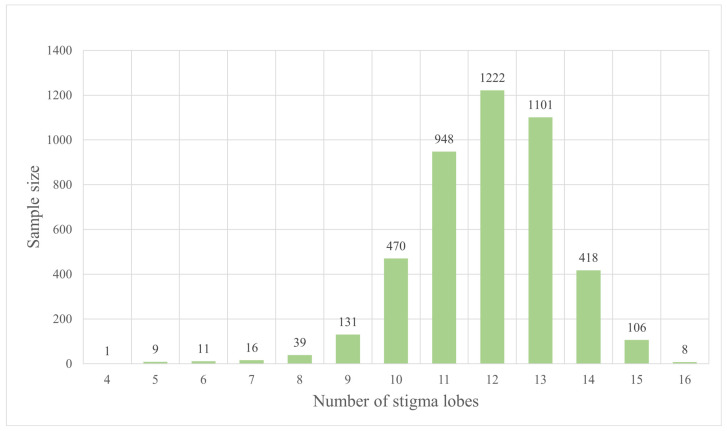
Number of stigma lobes per flower of *Papaver somniferum*.

**Figure 6 plants-12-02105-f006:**
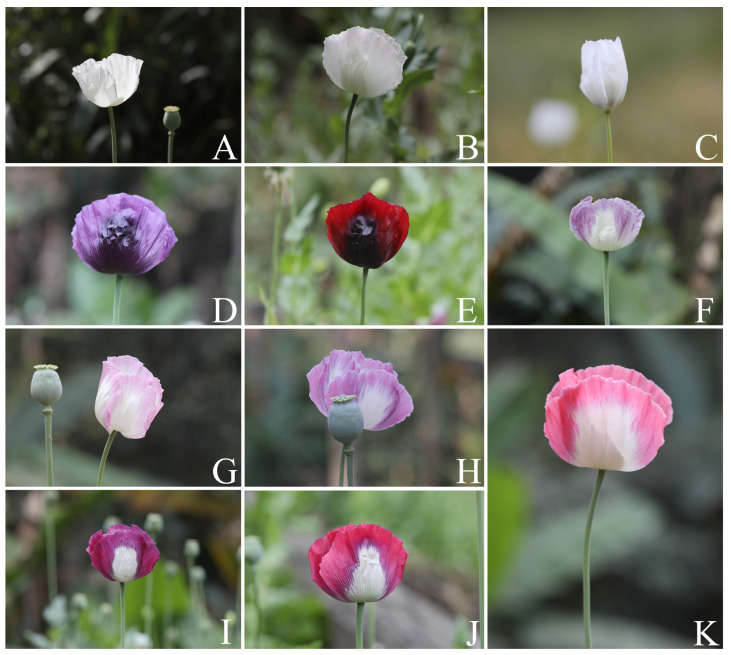
Variability in the color of the flowers of *Papaver somniferum*, Group 1, undulated petal margin. (**A**) White; (**B**) white tinged with pale pink on the apical part; (**C**) white with longitudinal purple lines on the basal part; (**D**) purple with a very dark purple middle blotch; (**E**) dark red (crimson) with a very dark purple middle blotch; (**F**) white–pale purple; (**G**) white–pale pink; (**H**) white–pale purple tinged with scattered pink on the pale purple part; (**I**) white–magenta (red–purple) tinged with purple on the basal part or the magenta basal part; (**J**) white–red tinged with purple on the basal part or the red basal part; and (**K**) white–pink tinged with purple on the basal part or the pink basal part. Photos: Chatchai Ngernsaengsaruay.

**Figure 7 plants-12-02105-f007:**
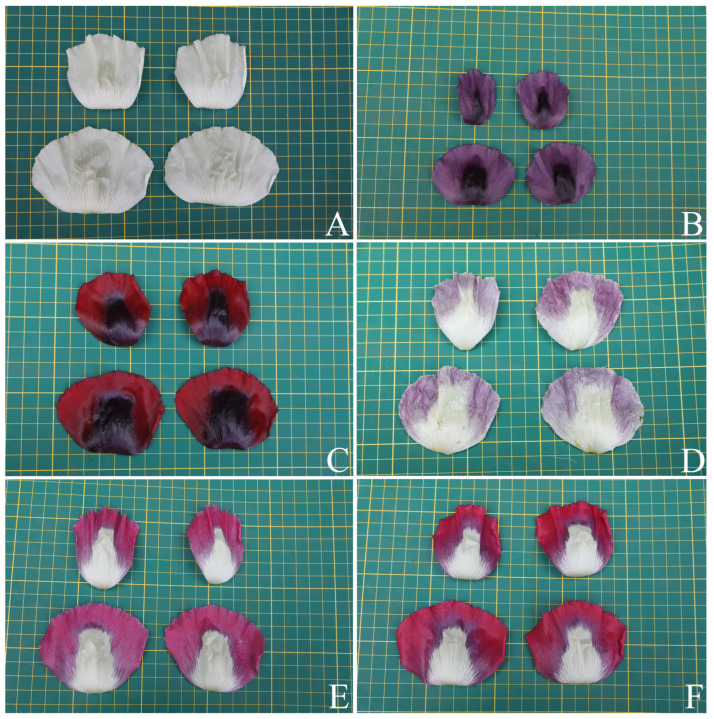
Variability in the color of the petals of *Papaver somniferum*, Group 1, undulated petal margin. (**A**) White; (**B**) purple with a very dark purple middle blotch; (**C**) dark red with a very dark purple middle blotch; (**D**) white–pale purple tinged with scattered pink on the pale purple part; (**E**) white–magenta tinged with purple on the basal part or the magenta basal part; and (**F**) white–red tinged with purple on the basal part or the red basal part. Photos: Chatchai Ngernsaengsaruay.

**Figure 8 plants-12-02105-f008:**
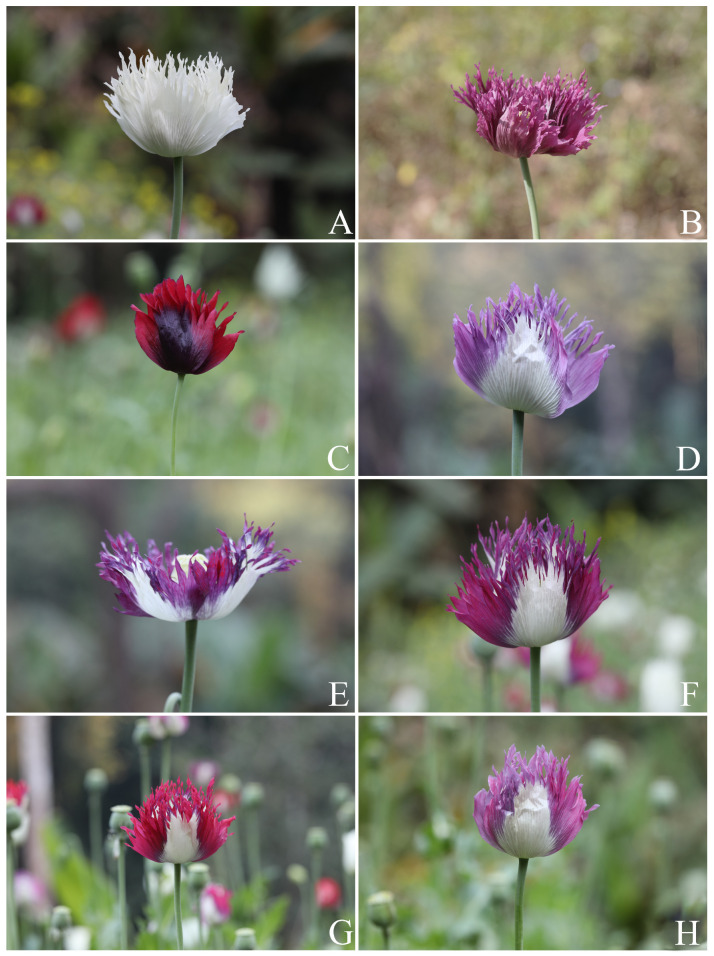
Variability in the color of the flowers of *Papaver somniferum*, Group 2, irregularly, longitudinally deeply lobed petal. (**A**) White; (**B**) purple with a very dark purple middle blotch; (**C**) dark red with a very dark purple middle blotch; (**D**) white–pale purple tinged with pink on the lobes or the pale purple basal part; (**E**) white–purple tinged with red on the lobes or the purple basal part; (**F**) white–magenta tinged with purple on the lobes or the magenta basal part; (**G**) white–red tinged with purple on the lobes or the red basal part; and (**H**) white–pink tinged with purple on the lobes or the pink basal part. Photos: Chatchai Ngernsaengsaruay.

**Figure 9 plants-12-02105-f009:**
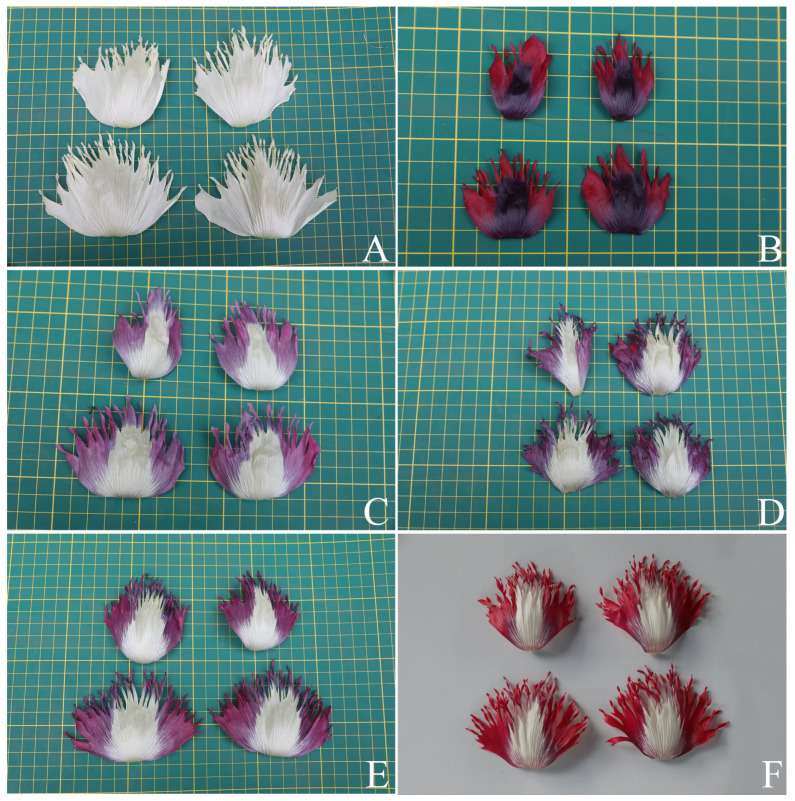
Variability in the color of the petals of *Papaver somniferum*, Group 2, irregularly, longitudinally deeply lobed petal. (**A**) White; (**B**) dark red with a very dark purple middle blotch; (**C**) white–pale purple tinged with pink on the lobes or the pale purple basal part; (**D**) white–purple tinged with red on the lobes or the purple basal part; (**E**) white–magenta tinged with purple on the lobes or the magenta basal part; and (**F**) white–red tinged with purple on the lobes or the red basal part. Photos: Chatchai Ngernsaengsaruay.

*Description*: Herb, annual, 0.27–1.5 m tall; latex, smelly, milky, turning pale orange or pale orange–pink, brown, and turning dark brown or black when dry (when exposed to the air), secreted from cut roots, stems, branches, leaves, peduncles, ovaries, and fruits. *Taproots* gradually narrow towards the apex, 4.1–23.7 cm long, 0.4–2 cm in diameter. *Stems* erect, branched or unbranched, cylindrical, basal part 0.3–1.7 cm in diameter, middle part 1.5–9.7 mm in diameter, glabrous, and glaucous (covered with a whitish waxy coating). *Leaves* simple, cauline, spiral, sessile; lamina unlobed or pinnately lobed, lanceolate, lanceolate-ovate, ovate, or broadly ovate, 3.2–37 × 1.5–17.5 cm, apex acute, base amplexicaul, margin from irregularly shallow to deeply dentate and undulated, green above, paler below, rarely purple–green, glabrous and glaucous on both surfaces, midrib broadly grooved (from the base to the middle) and flattened (from the middle to the apex) above, raised below, secondary veins irregularly branched, curvy, and connected in distinct loops (anastomosing) on both sides of the midrib, reticulated veinlets, distinct veins on both surfaces; fresh leaves crispy when crushed; chartaceous dry leaves; young plants with basal leaves, petiolate. *Flowers* solitary, terminal, 1–3(–6) per plant, showy, actinomorphic, fully opened and erect flowers, 2.9–10 × 2.1–10 cm; peduncle terete, 5–32 cm long, basal part 1–6.5 mm in diameter, middle part 0.8–6 mm in diameter, glabrous, glaucous; flower buds drooping at first, erect before anthesis, ellipsoid, lanceoloid-ovoid, or lanceoloid, 1.8–4.7 cm long, 0.7–2 cm in diameter, apex obtuse; two sepals, united in flower buds, nearly opened flowers separated from the base to the apex, caducous (sepals falling off before being fully opened flowers), green outside, turning pale green or pale green–yellow (nearly opened flowers), white inside, concave, elliptic, lanceolate-ovate, ovate, broadly elliptic, or lanceolate, 1.9–4.6 × 0.6–3.8 cm, apex obtuse, white membranous margin, both sides are glabrous, veins are dichotomously branched, distinctly inside; four petals, free, in two whorls, the outer pair larger than the inner pair, variable in color and morphological characteristics (Table 1, Figure 6, Figure 7, Figure 8 and Figure 9), caducous, crumpled in flower buds, the outer pair flabellate, 2.5–9.5 × 2.4–12 cm, the inner pair flabellate or obovate, 2.5–9.2 × 1.5–9.7 cm, margin undulated or irregularly and longitudinally deeply lobed, veins dichotomously branched; numerous stamens, (25–)60–223, usually longer than the pistil, 0.9–3 cm long; white or purple filaments with a white basal part, flattened, linear, 0.8–2.4 cm long, apex obtuse, the apical part broader than the lower part, with a central, longitudinal vein; pale yellow or cream colored (creamy white) anthers, turning brown, basifixed, narrowly oblong or oblong, sometimes linear or broadly elliptic, 1–7.2 × c. 1 mm, retuse at both ends, each side with a central, longitudinal groove, two-locular, opening by slits; pistil syncarpous, (4–)8–15(–16)-carpellate; short gynophore (a stalk carrying the pistil), 1.4–5 mm long, constricted; ovary superior, pale green, ovoid, lanceoloid-ovoid, ellipsoid, broadly ovoid, narrowly ellipsoid, or obovoid, 0.8–3.2 cm long, 0.4–2.1 mm in diameter, glabrous, glaucous, unilocular with parietal placentation (through intrusion of the radiate placentas to the center of the ovary but not fused), numerous ovules; sessile stigmas, pale green, pale yellow, or pale green with purple ridged stigmas, radiated, deeply (4–)8–15(–16)-lobed, ridged, apex obtuse, apical part to middle part of ridged stigmas grooved, united into flattened disk (a flat plat-shaped), recurved, covering the apical part of the ovary, umbrella-like, 0.4–2.1 cm in diameter; disk margin radiated, shallow (4–)8–15(–16)-lobed, rounded apex. *Fruits* capsular, poricidal (pores below stigmatic disk), green, glabrous, glaucous, turning stramineous, brown, or black–brown when dry, broadly ovoid, lanceoloid-ovoid, ovoid, ellipsoid, narrowly ellipsoid, or obovoid, 1.7–7 cm long, 0.8–3.7 cm in diameter; enlarged, persistent, sessile stigmatic disk, patent, 0.9–2.5 cm in diam., with (4–)8–15(–16) radiating stigmatic rays; stipitate 0.3–1.2 cm long, constricted; fruit stalks same as peduncles (of flowers). *Seeds* are numerous, 593–4685 seeds per capsule, white when young, turning creamy white, brown, black–brown, or black–grey, reniformed (kidney-shaped), 0.9–1.2 × 0.7–1 mm, faveolated, and have a minutely pitted surface (with small depressions) (as observed under a stereo microscope).

Measurements of the vegetative and reproductive parts of *P. somniferum* are presented in Table 2.

*Papaver somniferum* is related to *P. rhoeas* L. (common poppy, corn poppy, field poppy, flanders poppy, and red poppy), but differs in having a glabrous and glaucous herb [vs. with setose hairs and non-glaucous]; unlobed or pinnatifid leaves and an amplexicaul base [vs. pinnatisect or pinnatipartite leaves, not an amplexicaul base]; variable in color of petals (Table 1) (vs. deep scarlet, bright red, purple or bluish petals, occasionally a white margin, often with a dark spot at the petal base); and white filaments, occasionally purple with a white basal part, broadened at the apical part (Table 1) (vs. purple, not widened at the apical part). The characteristics of *P. rhoeas* were taken from Backer and Bakhuizen van den Brink [5], Long [6], Kiger and Murray [7], Zhang and Grey-Wilson [8], Egan et al. [9], CABI [29], and Missouri Botanical Garden [30].

*Distribution*: The species is distributed in various parts of southern Europe and Northern Africa (Macaronesia, Western and Central Mediterranean). Southern Europe: France, Italy, Spain, and Portugal, including the Mediterranean Sea; Corsica (Corse) under the French jurisdiction; Sardinia (Sardegna) and Sicily (Sicilia) under Italy; and the Balearic Islands (Baleares) under Spain; Macaronesia in the north Atlantic Ocean; Madeira Archipelago, governed by Portugal and the Canary Islands under Spain. Northern Africa: Algeria, Tunisia, Morocco, and Libya [3]. According to the International Narcotics Board [INCB] [15] and Carlin et al. [26], *Papaver somniferum* is an annual crop cultivated worldwide, but is legitimately grown by the pharmaceutical and food industries in Canada, Central and Southern America, Poland, the Czech Republic and Slovakia (the former Czechoslovakia), Hungary, Romania, the former Yugoslavia, Holland, France, Spain, Türkiye, Iran, India, and Australia.

*Cultivation Areas in Thailand*: Northern—Mae Hong Son [Khun Yuam (Mae Ukho subdistr.); Pai (Mae Hi, Mae Na Toeng, Pong Sa, and Wiang Nuea subdistr.)]; Chiang Mai [Chai Prakan (Si Dong Yen subdistr.); Chiang Dao (Chiang Dao, Mae Na, Mueang Khong, Mueang Ngai, and Thung Khao Phuang subdistr.); Mae Chaem (Ban Thap, Mae Na Chon, and Mae Suek subdistr.); Mae Taeng (Kuet Chang and Pa Pae subdistr.); Mae Wang (Mae Win subdistr.); Omkoi (Mae Tuen, Na Kian, Sop Khong, and Yang Piang subdistr.); Phrao (Pa Nai and San Sai subdistr.); Samoeng (Mae Sap subdistr.); and Wiang Haeng (Mueang Haeng subdistr.)]; Chiang Rai [Mae Suai (Pa Daet, Si Thoi, and Tha Ko subdistr.); Wiang Pa Pao (Mae Chedi Mai, San Sali, and Wiang subdistr.)]; Nan [Mae Charim (Nong Daeng subdistr.); Mueang Nan (Rueang and Sanian subdistr.); Na Noi (Santha subdistr.); Tha Wang Pha (Pa Kha and Si Phum subdistr.); Thung Chang (Lae, Ngop, and Thung Chang subdistr.); Wiang Sa (Mae Khaning and Yap Hua Na subdistr.)]; Lampang [Mueang Pan (Hua Mueang subdistr.)]; Phrae [Rong Kwang (Ban Wiang subdistr.); Song (Tao Pun subdistr.)]; Tak [Ban Tak (Thong Fa subdistr.); Mae Ramat (Khane Chue, Mae Tuen, and Sam Muen subdistr.); Phop Phra (Khiri Rat subdistr.); Sam Ngao (Ban Na subdistr.); Tha Song Yang (Mae Song, Mae Tan, Mae La, and Mae U-su subdistr.); Umphang (Mokro subdistr.); Wang Chao (Chiang Thong subdistr.)]; Phitsanulok [Nakhon Thai (Na Bua and Noen Phoem subdistr.)]; and Kamphaeng Phet [Khlong Lan (Pong Nam Ron subdistr.)]. Northeastern—Phetchabun [Lom Kao (Ban Noen and Wang Ban subdistr.)] (from Narcotics Crop Survey and Control Institute, ONCB′s observations, 2020–2023) (Figure 10).

*Habitat*: It is mostly illegally cultivated along the slopes of the hilly open areas of disturbed lower montane and lower montane pine-oak forests, at elevations of 800–1970 m above mean sea level (amsl). Such cultivations can sometimes be found in the open areas of disturbed mixed deciduous and dry evergreen forests along the streams of valleys, at lower elevations of 300–700 m amsl. 

*Pollinator*: The black dwarf honey bee, *Apis andreniformis* Smith, 1858, is known as “phueng man (ผึ้งม้าน)” in Thai (identified by Dr Chawatat Thanoosing using online database [31]).

*Specimens Examined*: Thailand. Northern—Mae Hong Son [Ban Mae Ya Noi, Mae Hi subdistr., Pai distr., illegally cultivated in an open area of a disturbed lower montane forest, 19°13′32.1″ N, 98°32′02.2″ E, 1130 m alt., flowers and fruits, 4 March 2023, *C. Ngernsaengsaruay* et al. *Ps05-04032023* (BK, BKF, and QBG); ibid., *C. Ngernsaengsaruay* et al. *Ps06-04032023* (BK, BKF, and QBG)], Chiang Mai [Ban Nam Ru, Mueang Khong subdistr., Chiang Dao distr., illegally cultivated in an open area of a disturbed lower montane forest, 19°25′14.6″ N, 98°36′43.2″ E, 1330 m alt., flowers and fruits, 6 January 2023, *C. Ngernsaengsaruay* et al. *Ps01-06012023* (BK, BKF, and QBG); ibid., *C. Ngernsaengsaruay* et al. *Ps02-06012023* (BK, BKF, and QBG); ibid., 19°25′59.6″ N, 98°36′54.4″ E, 1370 m alt., flowers and fruits, 6 January 2023, *C. Ngernsaengsaruay* et al. *Ps03-06012023* (BK, BKF, and QBG); ibid., *C. Ngernsaengsaruay* et al. *Ps04-06012023* (BK, BKF, and QBG); Ban Pa Lo, Mae Na subdistr., Chiang Dao distr., illegally cultivated in an open area near a disturbed lower montane pine-oak forest, 19°18′07.9″ N, 98°50′17.7″ E, 1510 m alt., flowers and fruits, 15 February 2023 (C. Ngernsaengsaruay et al.’s own observation, with photos); upper slopes of Doi Chang mountain, from 1260 m alt. to the summit (1765 m alt.), flowers, 11 January 1922, *J. F. Rock 1797* (E [E01030060]); Ban Li Su, Doi Chiang Dao, flowers, 19 July 1958, *Khantchai 944* (BKF); Doi Chiang Dao, flowers, 21 November 1962, *Khantchai 1216* (BKF); ibid., *Khantchai 1217* (BKF); Doi Chiang Dao, Chiang Dao distr., flowers, 14 January 1973, *S. Sutheesorn 2254* (BK); above Huai Hang Village (Li So), Doi Chiang Dao Wildlife Sanctuary, Chiang Dao distr., cultivated in an open field with a rugged limestone terrain, 1300 m alt., flowers, 2 February 1996, *J. F. Maxwell 96-173* (BKF and L [L4186776]); Pang Bo Hill, 1050 m alt., flowers and fruits, 11 March 1965, *T. Smitinand 8712* (L [L1824310]); Doi Ang Khang, Fang distr., 1600 m alt., flowers and fruits, 8 November 1973, *J. Sadakorn 300* (BK); ibid., 1500 m alt., flowers, 13 January 1975, *J. Sadakorn 574* (BK); Doi Nong Hoi, 1500 m alt., flowers, 17 July 1974, *Umpai 515* (BK); Chang Khian, Doi Suthep, Mueang Chiang Mai distr., 1400 m alt., flowers and fruits, 9 January 1988, *J. F. Maxwell 88-16* (L [L1824299]); Ban Kong Hae, Pong Yaeng subdistr., Mae Rim distr., 700–900 m alt., flowers, 12 December 1978, *Bjørnland and Schumacher 562* (BKF); Ban Pha Nok Kok, Pong Yaeng subdistr., Mae Rim distr., 1008 m alt., flowers and fruits, 20 January 2008, *K. Jatupol 08-186* (QBG); Bok Jan, c. 800 m alt., flowers and fruits, 7 February 1996, *W. Nanakorn* et al. *5949* (QBG); Mae Chon Luang, Mae Chaem distr., 1600 m alt., flowers, 16 July 1981, *Boonnak and Thani 3279* (BKF); Ban Sau Daeng, Mae Chaem distr., an open field, cultivated in pine forest, 1200 m alt., flowers, 24 January 1993, *J. F. Maxwell 93-80* (BKF and L [L4186851])]; Phayao [Doi Phu Langka, flowers, 21 September 1938, *Unknown 265* (BKF99004); ibid., *Unknown s.n.* (BKF99005)], Phrae [location not specified, 1500 m alt., flowers, s.d., *Unknown s.n.* (BKF111); and ibid., *Unknown s.n.* (BKF112)], province not specified [location not specified, flowers, 29 November 1933, *H. B. Garrett 843* (P [P02308256]), *H. B. Garrett 844* (E [E01030061]), and *H. B. Garrett 845* (P [P02308255])]. 

Türkiye. *P. Sintenis 836* (K [K000653144]); *P. Sintenis 4202* (P [P02308253]); and *B. V. D. Post s.n.* (E [E00403610]). 

Isarael. Jerusalem, *F. Meyers and J. E. Dinsmore 4973* (E [E00403611] and L [L1824280]). 

India. *Wight′s Herbarium, East India Company Herbarium [EICH] 8118A* (K-W [K001129004]); Bangalore (Bengaluru), *Madras Herbarium, EICH 8118B* (K [K001129005]); Calcutta Botanical Garden, *EICH 8118C* (K-W [K001129006]); Bankipore, *G. Watt 1104* (E [E01030122]); Manipur, *G. Watt 7208* (E [E01030120]); cultivated in Herb. Hort. Bot. Calcuttensis, *S. Kurz 735* (E [E01030121]); Punjab, *T. Thomson 956* (P [P02308249]); and Jammu and Kashmir, *Schlagintweit s.n.* (P [P02308260]). 

Nepal. *O. Polunin, W. R. Sykes, and L. H. J. Williams 1919* (E [E00914679]). 

Myanmar. Kachin, *J. Keenan, U. Tun Aung and U Tha Hla 3855* (E [E01030131]). 

China. Patung Hsien, *Ho-Chang Chow 267* (E [E01035975]); Chengtu, *S. S. Chien 5299* (E [E01035976] and P [P02308264]); Kwangsi, *A. N. Steward and H. C. Cheo 85* (P [P02308263]); Shaanxi, *Bélard 8* ([P [P02308269]); Sichuan, *J. Hopkingson 119* (P [P02308273]); cultivé Chine boréale, *von A. A. Bunge s.n.* (P [P03166473]); and Tibet, *H. J. Walton s.n.* (P [P02308251]). 

Japan. Japonia, *O. G. J. Mohnike s.n.* (L [L1824307]); and Japonia, *C. J. Textor s.n.* (L [L1824219]). 

Laos. Xiangkhouang, Xieng Khuang, *A. J. B. Chevalier 2297* (L [L4170238] and P [P02308254]); *Poilane 2364* (P [P02308250]). 

Philippines. Luzon, Mountain province, *F. C. Garcia PNH 34991* (L [L1824296]). 

Australia. Parramatta NSW., *K. L. Wilson 1953* (L [L1824235]); and Wagga distr. NSW., *J. Sutherland 7864* (L [L1824297]). 

America. USA. California, San Francisco, *S. Kawahara and F. Yamasaki 1389* [E01030117]); Mexico. Puebla, *G. J. B. Arsène 216* (L [L1824298]), and Ecuador. Pichincha, *Rivet s.n.* (P [P02308286]). 

Africa. Algeria. *Davis 51732* (E [E01035914]); Alger, *Hohenacker s.n.* (P [P03166530]); Morocco. *J. J. F. E. de Wilde, P. A. W. J. de Wilde and J. Dorgelo 2949* (L [AMD82636, L1824281, and WAG1146734]); Ethiopia. *P. C. M. Jansen 391* (L [WAG1146722]); and Mauritius. *P. Commerson* (P [P02338563 and P02338564]). 

Europe. Sweden. Östergötlands län, *H. Mosén* (P [P03166561]). 

United Kingdom. Kew Gardens, England, *T. A. Cope RBG 386* (K [K000914023]); Lincolnshire, England, *E. A. Woodruffe-Peacock s.n.* (BM [BM013414465]); Scotland, *A. Craig-Christie s.n.* (E [E00761590 and E00761621]); Scotland, *W. E. Evans Herbarium s.n.* (E [E00761601]); Scotland, *Barkoudah, Ferguson and Brummitt s.n.* (E [E00761610]); Scotland, *D. K. Kevan s.n.* (E [E00851026]); Edinburgh Granton, Scotland, *O. M. Stewart 276* (E [E00761612]). 

Germany. Göttingen, *A. C. W. Staring s.n.* (L [WAG1146721]); Württemberg, *R. F. Hohenacker 600* (P [P03166527]); and Sachsen, *Krieger s.n.* (P [P02681829]). 

Netherlands. Verwilderd, duinen, Katwijk, *W. J. C. Kooper s.n.* (L [U1467913]); and Neerlandica, Amersfoort, Verwilderd in tuin, *R. C. Bakhuizen van den Brink 4548* (L [U1467912]). 

Belgium. Belgica, Vermoedelijk verwilderd, Chevetogne, *W. H. A. M. Hekking 697* (L [U1467778]). 

Austria. Nockberge, Südöstlich von Bad Kleinkirchheim, *B. Zimmer 2573* (L [WAG1146720]). 

Switzerland. Germanicae et Helveticae, *W. D. J. Koch s.n.* (L [L1824295]). 

Italy. Potenza, *N. Gavioli (Hb. J. Arènes no 00458)* (P [P02681828]); and Pisa, *P. Savi s.n.* (P [P03166554]). 

Greece. Levant. Schraf-Mohn, *H. H. s.n.* (L [L1824279]). 

Cyprus. Kalokhorio to Athrakos, *J. R. Edmondson and M. A. S. McClintock 2974* (E [E00403613]). 

France. Fréjus, *Perreymond s.n.* (K [K000653105]); location not specified, *Unknown s.n.* (K [K000653106]); Hohwald (Bas Blui) langs de weg, *A. Gorter 1038* (L [L1824278]); Maffliers, Environs du bois de Belloy-Montsoult, *Herbier de M. Paul Jovet, Reçu en 1991* (P [P00504473]); Torcy, *Herbier de M. Paul Jovet. Reçu en 1991* (P [P00504603]); Basse- Normandie, *P. Porte s.n.* (P [P02467196]); 34 Hérault, *A. Dubuis (Hb B. de Retz n° 78715)* (P [P02681479]); 77 Seine-et-Marne, *Hennecart s.n.* (P [P02470138]); Languedoc-Roussillon, *Tuzkiewicz s.n.* (P [P02558905]); cultivé Besançon, *J. C. M. Grenier* (P [P03166446]); Charenton, *Maire s.n.* (P [P03166450]); Picardie, *A. Camus and E. G. Camus* 146 (P [P03166451]); Les Croisettes, *Eloy de Vicq s.n.* (P [P03166609]); and Limousin, *H. Bouby 6038* (P [P03767267]). 

Spain. Insul. Canar. Fuerleventura, *M. Fleischer and E. Fleischer-Haighton 337* (L [U1467821]); Huesca, *S. Barrier 72294A* (P [P02467194]); and Canary Islands, Gran Canaria, boven Talsequillo, *C. H. Andreas and H. D. Schotsman s.n.* (L [WAG1146731]). 

Portugal. Rij Faro, Z. Portugal, *A. J. G. H. Kostermans and W. Kruyt 742* (L [L1824287]); Algarve, *J. van Kasteel 4095* (L [L4170122]); Madeira, *G. Mandon s.n.* (P [P02682530]); Setúbal, *Welwitsch s.n.* (P [P03166474]); and Faro, *E. Bourgeau s.n.* [P03167520]). 

*IUCN Conservation Status*: Least Concern (LC) in agreement with Chadburn [32]. 

*Growing Period and Phenology*: Based on the observations of Fin (opium poppy) in Thailand, its growing period can be classified into three groups: (1) Early fin, also called “fin do (ฝิ่นดอ)”, spans three months from August to October and is characterized as the rainy season; (2) Winter fin, also known as “fin pi (ฝิ่นปี)”, spans six months from October to February and is characterized as the winter season; and (3) Summer fin, also known as “fin rue du ron (ฝิ่นฤดูร้อน)”, spans three months from February to April and is characterized as the summer season. The length of the growing season for the crop is about 110 days. Within this period, the flower bud initiation before anthesis (before the flower is fully open) takes place 75–77 days after sowing, with another 3–5 days for the flower to open (78–82 days), and another 8–10 days for the fruiting stage to set in (about 85 days). Latex collection is conducted from 88 to 93 days after sowing, while the seeds are harvested after 110 days of sowing. According to Schiff [23], *Papaver somniferum* is harvested for its latex 5–10 days after the flowering petals have fallen from the plant.

*Etymology*: *Papaver somniferum* was named by Carl Linnaeus (1707–1778), a Swedish botanist, physician, and zoologist [33,34]. The generic name “*Papaver*” comes from a Latin name for poppies, including the opium poppy [35,36]. The specific epithet “*somniferum*” is a Latin word meaning sleep-inducing [36], referring to the latex exuded from the fruit, which has a sedative property of some opiate drugs. 

*Vernacular Name*: Fin (ฝิ่น) (General); Breadseed poppy and Opium poppy (English). 

*Uses*: Opiate drugs derived from the milky latex found in unripe fruits (capsules) of opium poppy, which include morphine, heroin, codeine, and other alkaloids, are used in making analgesics [2,8]. Poppy seeds (mostly *Papaver somniferum* and *P. rhoeas*) are commonly used as a condiment and are a traditional ingredient in breads, bagels, and other baked goods. A paste of ground poppy seeds and sugar is used in making central and eastern European patisserie [2]. The seeds are used in food, oils, and pharmaceuticals to treat inflammation, and are cardiotonic, mildly astringent, analgesic, and sedatives [25]. 

Opium alkaloids (morphine, codeine, thebaine, noscapine, and papaverine) have been detected in poppy seeds and are widely used by the food industry for decoration and flavoring, but can introduce opium alkaloids into the food chain. Of the opium alkaloids found in poppy seeds, morphine and codeine are the most pharmacologically active compounds. The European Food Safety Authority has set an acute reference dose of 10 µg/kg of body weight as a safe level for morphine consumed through food products [26].

### 2.2. Stem and Leaf Lamina Anatomy

#### 2.2.1. Stem Anatomy

Transverse sections of the stem of *Papaver somniferum* are circular outlines and consist of the epidermis, cortex, vascular bundles, and pith. The epidermal cells are covered by the cuticle. Three types of distinct tissues are found in the cortex: parenchyma, sclerenchyma, and collenchyma. Several collenchyma layers are found below the epidermis, and several sclerenchyma layers are found below the collenchyma layers. The central pith is made up of parenchyma cells. Two rings of discontinuous and widely spaced, numerous collateral vascular bundles are recognized. Laticifers are associated with the phloem tissues of the vascular bundles (Figure 11).

#### 2.2.2. Leaf Lamina Anatomy

*Leaf lamina epidermis*: The leaf lamina epidermis of *Papaver somniferum* is usually covered with wax. The epidermal cells on the adaxial surface are polygonal in shape and have straight or slightly curved anticlinal cell walls, while those on the abaxial surface are polygonal or irregular shapes and have straight, slightly curved, sinuate, or strongly sinuate anticlinal cell walls. The stomata are confined to the lower epidermis (hypostomatic leaves) and are anomocytic-type without subsidiary cells, surrounded by epidermal cells. The size of the stomata are 22.29–52.77 (with a mean of 37.50 ± 6.75) µm length, 8.11–28.43 (with a mean of 15.92 ± 3.69) µm width, and the stomatal density is 54–199/mm^2^ (with a mean of 89.29 ± 24.97) (Figure 12). 

*Transverse section of leaf lamina*: The mesophyll is dorsiventral. It is composed of 1–3 layers of palisade cells and several layers of spongy cells, but not distinctly differentiated into palisade and spongy regions in agreement with Metcalfe and Chalk [37]. The vascular bundle is solitary and surrounded by parenchyma cells. The midrib can be distinguished by a larger vascular bundle than the other veins. Laticifers are found in the phloem areas (Figure 12). 

### 2.3. Palynology

The pollen grains of *Papaver somniferum* are monads, isopolar, and radially symmetrical. The shape of pollens can be spheroidal or prolate spheroidal, and sometimes oblate spheroidal [P/E ratio = 0.99–1.12 (with a mean of 1.03 ± 0.03)]; the polar axis (P) diameter is 23.20–38.55 (with a mean of 31.14 ± 3.35) µm and the equatorial axis (E) diameter is 21.46–35.03 (with a mean of 30.20 ± 3.33) µm, which are medium-sized and sometimes small-sized. The outline shape in polar view (amb) is circular, and the amb diameter is 21.41–37.40 (with a mean of 30.65 ± 3.35) µm. The pollen aperture is tricolpate and the pore is fusiform, 12.83–31.15 (with a mean of 21.77 ± 3.96) µm length, 8.36–15.18 (with a mean of 11.06 ± 1.86) µm width. The exine thickness is 1.24–2.81 (with a mean of 2.02 ± 0.39) µm and the sculpturing is microechinate (Figure 13).

## 3. Discussion

According to previous studies, the species can be an herb without hairs (glabrous) or with sparsely setose hairs [5,7,8,9,10,11,22]; however, from our observations, we only found the plant without hairs, in agreement with Hooker and Thomson [4] and Long [6].

According to previous studies, the latex exuded from the fruit of *Papaver* is usually white (milky), sometimes watery, yellow, orange, or red [4,6,7,8,9,10,11], but from our observations, we found the latex can be smelly, milky, turning pale orange or pale orange–pink, brown, and turning dark brown or black when dry.

According to Hooker and Thomson [4], Backer and Bakhuizen van den Brink [5], Long [6], Kiger and Murray [7], Zhang and Grey-Wilson [8], Jafri and Qaiser [9], Egan et al. [10], Suddee [11], and India Biodiversity Portal [22], the arrangement, shape, and size of leaves are alternate, ovate, oblong, ovate-oblong, linear-oblong, broadly lanceolate, or obovate, and 5–25(–30) × 2–15 cm; however, from our observations, we found that the leaves can be spiral, lanceolate, lanceolate-ovate, ovate, or broadly ovate, and sometimes larger, with a range of 3.2–37 × 1.5–17.5 cm.; the shape, size, and number of persistent, radiate stigmas of capsules are globose, subglobose, globose-ovoid, globose-broadly ovoid, oblong-ellipsoid, broadly ellipsoid, or urceolate, 2–7(–9) × 2.5–6(–7.5) cm and 5–15(–18) radiate stigmas; furthermore, from our observations, we found the capsules can be broadly ovoid, lanceoloid-ovoid, ovoid, ellipsoid, narrowly ellipsoid, or obovoid, and sometimes smaller, with a range of 1.7–7 cm long, 0.8–3.7 cm in diameter, and stigmas with (4–)8–15(–16) radiating lobes.

Previously, the anomocytic (irregular-celled) type was termed ranunculaceous, which was taken from the family name Ranunculaceae, in which it was first observed [37].

From our observations, the shape and anticlinal walls of the epidermal cells on both leaf surfaces of *Papaver somniferum* are similar to those of *P. bracteatum* Lindl. and *P. orientale* L. from the result of Tavakkoli and Assadi [38].

According to Metcalfe and Chalke [37], the stems in the transverse section of *Papaver* (e.g., *P. orientale*) sometimes have several rings of collateral vascular bundles; furthermore, we found two rings of discontinuous and widely spaced collateral vascular bundles.

According to Bird et al. [27], the benzylisoquinoline alkaloids of opium poppy accumulate in the cytoplasm, or latex, of specialized laticifers that accompany vascular tissues throughout the plant; however, from our study, we observed them in stems and leaves.

According to Özkök and Sorkun [28], the pollen grains of *Papaver somniferum* are: white and purple opium poppy flowers that are similar in morphology; tricolpate in aperture, oblate spheroidal in shape (P/E ratio = 0.95), and microechinate in sculpturing. Furthermore, from our observations, we found that the shape of pollens are spheroidal or prolate spheroidal (P/E ratio = 0.99–1.12).

According to Özkök and Sorkun [28], *Papaver somniferum* white flower pollen (P diameter with a mean of 27.78 ± 1.04 µm; E diameter with a mean of 29.04 ± 0.82 µm; amb diameter with a mean of 29.02 ± 0.73 µm; colpus length = 22.78 µm; colpus width = 8.34 µm; and exine thickness = 1.00 µm) and *P. somniferum* purple flower pollen (P diameter with a mean of 28.76 ± 1.03 µm; E diameter with a mean of 30.28 ± 0.94 µm; amb diameter with a mean of 29.98 ± 1.29 µm; colpus length = 23.78 µm; colpus width = 9.90 µm; and exine thickness = 1.00 µm); however, from our observations, we found very slight differences in these characters.

## 4. Materials and Methods

Plant specimens of *Papaver somniferum* were observed and collected in the northern region of Thailand (Mae Hong Son and Chiang Mai provinces) (Figure A1). Herbarium specimens deposited in BK, BKF, QBG, and those included in the digital herbarium databases of BM, E, K, K-W, L, and P were examined by consulting the taxonomic literature [4,5,6,7,8,9,10,11,22,33,39] (acronyms follow those in the study by Thiers [40]). The herbarium accession number can be seen on the specimens examined. The taxonomic history of this species was compiled using the taxonomic literature and online databases POWO [3] and IPNI [41]. The morphological characteristics, cultivation areas, habitats, pollinators, growing periods, phenology, vernacular names, and uses were described from our observations during field work and from label information on the specimens examined.

The preparation of plant samples was for anatomical observation. Transverse sections of the stems and the leaf lamina were through the midribs. The stem and leaf samples were dehydrated in an increasing ethanol concentration series of 30%, 50%, 70%, 95%, and absolute ethanol, embedded in paraffin, sectioned with a rotary microtome at 16–20 µm thickness with Haupt′s adhesive affixing paraffin sections to slides, stained with safranin and fast green, cleared with xylene, and mounted in DePeX mounting media. Leaf epidermal preparations were made by peeling and mounting on slides. The anatomical characteristics were investigated and recorded photographically with an Olympus BX53 microscope and an Olympus DP74 microscope digital camera at the Department of Botany, Faculty of Science, Kasetsart University (KU). The anatomical terminologies follow those in the study by Metcalfe and Chalk [37]. 

The samples of pollen grains were taken from the herbarium specimen collected from Mae Hong Son and Chiang Mai provinces (*C. Ngernsaengsaruay* et al. *Ps01-06012023*, *C. Ngernsaengsaruay* et al. *Ps02-06012023*, *C. Ngernsaengsaruay* et al. *Ps03-06012023*, *C. Ngernsaengsaruay* et al. *Ps04-06012023*, *C. Ngernsaengsaruay* et al. *Ps05-04032023*, and *C. Ngernsaengsaruay* et al. *Ps06-04032023*). They were examined and recorded photographically with an Olympus BX53 microscope and an Olympus DP74 microscope digital camera. Materials were prepared for scanning electron microscopy (SEM) at the Scientific Equipment Centre, Faculty of Science, KU by mounting pollen grains on stubs using double-sided sellotape, sputter-coating them with gold and examining them using an FEI Quanta 450 SEM (Hillsboro, OR, USA) at 15.00 KV. The characteristics of fifty pollen grains (polarity, symmetry, shape, size, aperture, exine thickness, and sculpturing) were examined and measured, following Erdtman [42,43] and Simpson [44]. The pollen morphology terminologies follow those of Punt et al. [45].

## 5. Conclusions

Opium poppy is illegal in most parts of the world, including Thailand; as such, information about opium poppy is mostly lacking and its characteristics poorly known. We have obtained permission to study and support from the Narcotics Crop Survey and Control Institute, ONCB, Ministry of Justice. In this research project, we provide the morphology, taxonomy, anatomy, and palynology of *Papaver somniferum*. As a result, we update detailed morphological descriptions for this species, which is more than the information in previous studies in Thailand and neighboring countries. With proven benefits of opium poppy already reported, the Office of the Narcotics Control Board, Thailand, plans to study opium poppy for further usage in the pharmaceutical industry.

The species can be characterized as a glabrous and glaucous herb, with unlobed or pinnately lobed leaves and an amplexicaul base; with variations in color and morphological characteristics of petals; and white filaments, occasionally purple with a white basal part, broadened at the apical part.

From our study, we found that the number of seeds per capsule of *Papaver somniferum* has a range of 593–4685. We realized that the number of seeds per capsule depends on the size of fruits and number of carpels.

Even when *Papaver somniferum* is notoriously known as a highly addictive non-synthetic narcotic, its usage in deriving pain killers (morphine) should not be ignored. The seeds are used in food, oils, and pharmaceuticals to treat inflammation, are cardiotonic, mildly astringent, analgesic, and sedatives. As per international law, the usage of *P. somniferum* should be strictly controlled and only used under supervision. However, a complete ban on the cultivation of *P. somniferum* would be counterproductive, given its high potential usage. Regarding the variation in the morphology of *P. somniferum* even in the same location, this might suggest that it can have various pharmaceutical properties.

## Figures and Tables

**Figure 10 plants-12-02105-f010:**
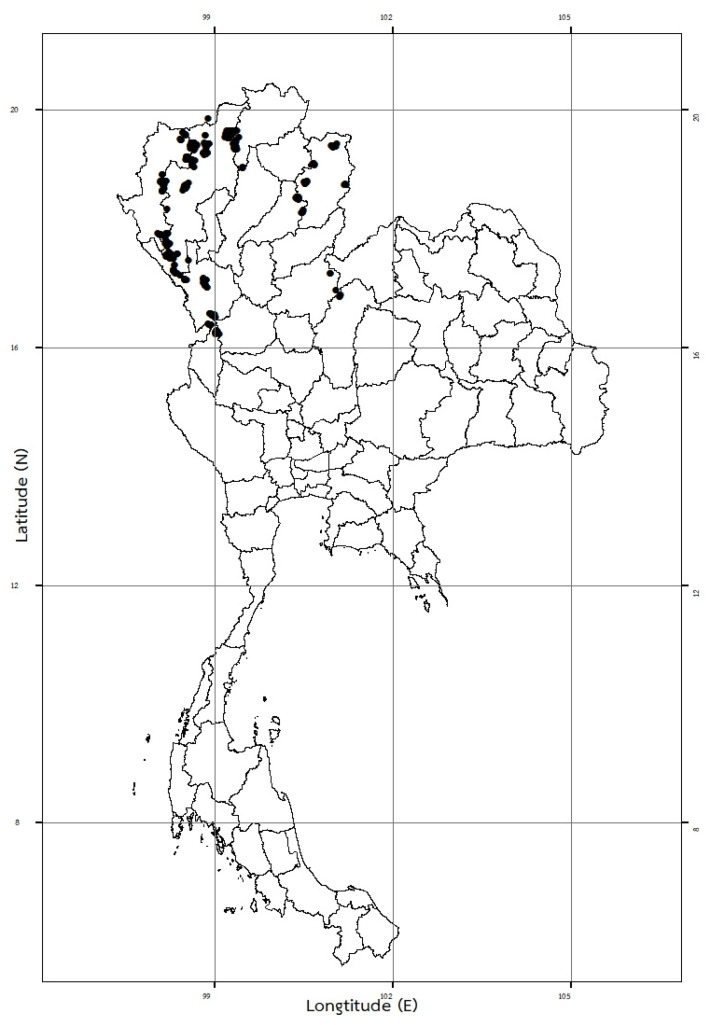
The illegal cultivation areas of *Papaver somniferum* in Thailand. It is cultivated in northern and northeastern Thailand (from Narcotics Crop Survey and Control Institute, ONCB′s observations, 2020–2023).

**Figure 11 plants-12-02105-f011:**
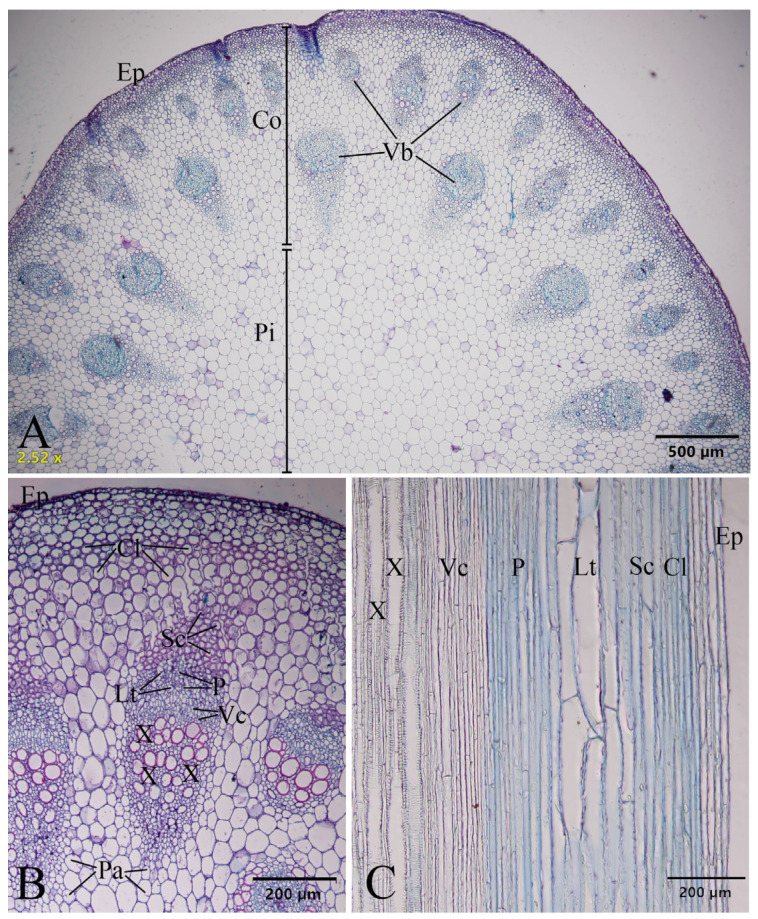
Stem anatomical characteristics of *Papaver somniferum*. (**A**,**B**) Transverse sections; and (**C**) longitudinal section. Cl: collenchyma, Co: cortex, Ep: epidermis, Lt: laticifer, P: phloem, Pa: parenchyma, Pi: pith, Sc: sclerenchyma, Vb: vascular bundle, Vc: vascular cambium, and X: xylem. Photos: Pichet Chanton.

**Figure 12 plants-12-02105-f012:**
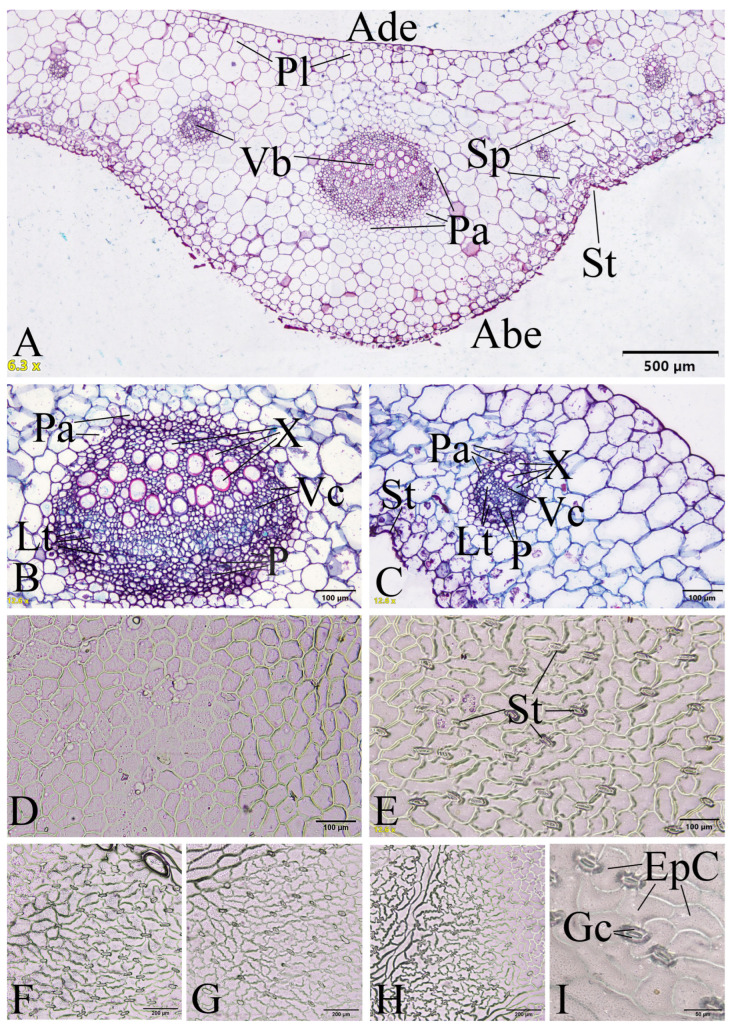
Leaf lamina anatomical characteristics of *Papaver somniferum*. (**A**–**C**) Transverse sections; (**D**) epidermal cells on the adaxial surface showing polygonal shapes; and (**E**–**I**) epidermal cells on the abaxial surface showing polygonal and irregular shapes. Ade: adaxial epidermis, Abe: abaxial epidermis, Epc: epidermal cell, Gc: guard cell, Lt: laticifer, P: phloem, Pa: parenchyma, Pl: palisade mesophyll, Sp: spongy mesophyll, St: stoma, Vb: vascular bundle, Vc: vascular cambium, and X: xylem. Photos: Pichet Chanton.

**Figure 13 plants-12-02105-f013:**
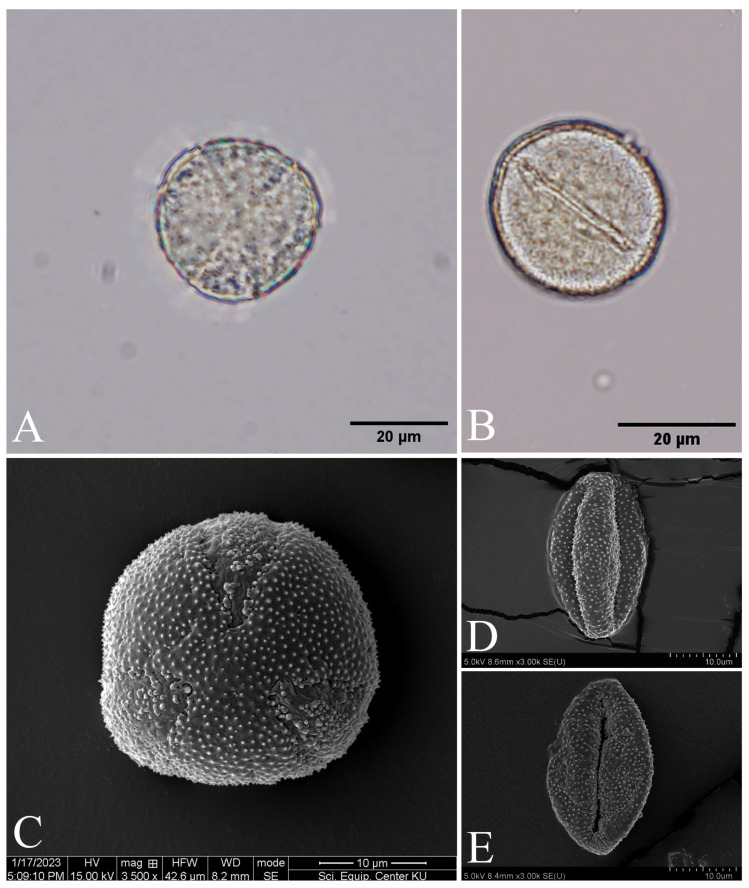
LM (**A**,**B**) and SEM (**C**–**E**) micrographs of pollen grains of *Papaver somniferum*. (**A**,**C**) Polar view; and (**B**,**D**) equatorial view. Materials from *C. Ngernsaengsaruay* et al. *Ps03-06012023* (BKF) and *C. Ngernsaengsaruay* et al. *Ps04-06012023* (BKF). Photos: Pichet Chanton (**A**,**B**); and Scientific Equipment Centre, Faculty of Science, Kasetsart University (**C**–**E**).

**Table 1 plants-12-02105-t001:** Variability in the flower color and morphological characteristics of *Papaver somniferum*.

Group 1		Group 2	
Undulated Petal Margin		Irregularly, Longitudinally Deeply Lobed Petal	
Petal Color	Filament Color	Petal Color	Filament Color
White	White	White	White
White tinged with pale pink on the apical part	White	–	–
White with longitudinal purple lines on the basal part	White	–	–
Purple with a very dark purple middle blotch	Purple with a white basal part	Purple with a very dark purple middle blotch	Purple with a white basal part
Dark red (crimson) with a very dark purple middle blotch	Purple with a white basal part	Dark red with a very dark purple middle blotch	Purple with a white basal part
White–pale purple	White	–	–
White–pale pink	White	–	–
White–pale purple tinged with scattered pink on the pale purple part	White	White–pale purple tinged with pink on the lobes or the pale purple basal part	White
–	–	White–purple tinged with red on the lobes or the purple basal part	White
White–magenta (red–purple) tinged with purple on the basal part or the magenta basal part	White	White–magenta tinged with purple on the lobes or the magenta basal part	White
White–red tinged with purple on the basal part or the red basal part	White	White–red tinged with purple on the lobes or the red basal part	White
White–pink tinged with purple on the basal part or the pink basal part	White	White–pink tinged with purple on the lobes or the pink basal part	White

**Table 2 plants-12-02105-t002:** Measurements of the vegetative and reproductive parts of *Papaver somniferum*.

Measurements of Morphological Characteristics	Units	Sample Size	Range	Mean ± SD
Stem height	cm	65	27.0–150.0	91.41 ± 29.96
Taproot length	cm	45	4.1–23.7	13.95 ± 4.05
Taproot diameter	cm	45	0.4–2.0	9.85 ± 3.70
Basal stem diameter	cm	55	0.3–1.7	0.89 ± 0.35
Middle stem diameter	mm	45	1.5–9.7	5.25 ± 2.02
Leaf lamina length	cm	275	3.2–37.0	16.43 ± 7.18
Leaf lamina width	cm	275	1.5–17.5	7.79 ± 3.46
Leaf lamina length/width ratio	ratio	275	1.1–5.6	2.29 ± 0.87
Number of flowers per plant	flowers	100	1–3(–6)	2.14 ± 1.11
Flower length	cm	90	2.9–10.0	6.96 ± 1.67
Flower width	cm	90	2.1–10.0	5.81 ± 1.61
Peduncle length	cm	90	5.0–32.0	19.82 ± 6.11
Basal peduncle diameter	mm	90	1.0–6.5	3.68± 1.36
Middle peduncle diameter	mm	90	0.8–6.0	3.26 ± 1.23
Flower bud length	cm	50	1.8–4.7	3.41 ± 0.68
Flower bud diameter	cm	40	0.7–2.0	1.47± 0.29
Flower bud length/diameter ratio	ratio	40	1.8–3.2	2.39± 0.31
Sepal length	cm	100	1.9–4.6	3.33 ± 0.65
Sepal width	cm	100	0.6–3.8	1.96 ± 0.84
Flower bud length/width ratio	ratio	100	1.1–4.0	1.91 ± 0.59
Outer pair petal length	cm	100	2.5–9.5	6.37 ± 1.87
Outer pair petal width	cm	100	2.4–12.0	7.39 ± 2.52
Inner pair petal length	cm	100	2.5–9.2	6.25 ± 1.83
Inner pair petal width	cm	100	1.5–9.7	5.69 ± 2.24
Number of stamens per flower	stamens	40	(25–)60–223	124.78 ± 53.51
Stamen length	cm	90	0.9–3.0	1.91 ± 0.56
Filament length	cm	90	0.8–2.4	1.53 ± 0.42
Anther length	mm	90	1.0–7.2	3.74 ± 1.57
Gynophore length	mm	50	1.4–5.0	3.38 ± 0.93
Ovary length	cm	60	0.8–3.2	1.89 ± 0.61
Ovary diameter	cm	60	0.4–2.1	1.15 ± 0.48
Ovary length/diameter ratio	ratio	60	1.2–3.2	1.73 ± 0.35
Number of stigma lobes	stigma lobes	4480	(4–)8–15(–16)	11.92 ± 1.48
Stigma diameter	cm	60	0.4–2.1	1.10 ± 0.37
Fruit length	cm	55	1.7–7.0	4.24 ± 1.39
Fruit diameter	cm	55	0.8–3.7	2.55 ± 0.81
Fruit length/diameter ratio	ratio	55	1.1–4.2	1.78 ± 0.65
Stigmatic disk diameter	cm	55	0.9–2.5	1.91 ± 0.39
Stipitate length	cm	65	0.3–1.2	0.63 ± 0.20
Seed length	mm	100	0.9–1.2	1.01 ± 0.07

## Data Availability

All relevant data can be found within the manuscript.

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
