# Peer review of "Morphology, Taxonomy, Anatomy, and Palynology of the Opium Poppy (Papaver somniferum L.) Cultivation in Northern Thailand"

_plants, 2023, doi:10.3390/plants12112105_

Round 1
Reviewer 1 Report
The introduction section is not well-written in the format as expected for the research paper to be published in this reputed Journal. I recommend re-write and expanding: Introduction section with updated literatures.
The manuscript objectives still need improvement and explanation in detail.
The authors are advised to improve the manuscript in terms of adequate language levels as well as research paper structure.
The authors should elaborate more on their findings and discussion compared to other studies, to their importance, as well as to plant profile in the screened region.
English should be improved; grammar needs enhancement in many sentences and paragraphs.
Figure 1, 5, and 10 are needed for resolution enhancement.
The discussion should show how these new findings support the research question and hypothesis.
Please check the References in-text and end-list for uniformity in style.
The introduction section is not well-written in the format as expected for the research paper to be published in this reputed Journal. I recommend re-write and expanding: Introduction section with updated literatures.
The manuscript objectives still need improvement and explanation in detail.
The authors are advised to improve the manuscript in terms of adequate language levels as well as research paper structure.
The authors should elaborate more on their findings and discussion compared to other studies, to their importance, as well as to plant profile in the screened region.
English should be improved; grammar needs enhancement in many sentences and paragraphs.
Figure 1, 5, and 10 are needed for resolution enhancement.
The discussion should show how these new findings support the research question and hypothesis.
Please check the References in-text and end-list for uniformity in style.
Author Response
Respond to Reviewer 1
The authors are grateful to thank the reviewer 1 for valuable comments.
Thank you very much for your kind consideration.
For improvements and corrections in this paper, the authors highlighted in red.
We add four paragraphs and nine references [12–20] in Introduction (Line 49–87).
Poppy is reported to have originated in ancient Mesopotamia (modern day Iraq and Kuwait) [12] and sourced for latex (milky sap) whose alkaloids are used in the pharmaceutical industry or for poppy seeds that are used in the food industry [13]. Poppy is illegal in most parts of the world including Thailand. Only a few countries that the legal cultivation of opium poppy for pharmaceutical and food industry is done in Australia, Canada, India, Central and Southern America, Türkiye (Turkey), Russia, Czech Republic, Slovakia, Holland, France, Hungary, Iran, Poland, Romania, and Spain [14,15].
Different cultivars with low alkaloid content can be used in the food industry for seed or oil production [16] such as in some European countries (e.g. Germany and the Czech Republic), which only permit the cultivation of ‘low‐morphine’ varieties of Papaver somniferum. Apart from these, some countries only grow varieties to derive pharmaceutical use from the capsules (e.g. Australia, France, and Spain) or for both culinary and pharmaceutical uses (e.g. Hungary and Slovakia). Poppy grows on a wide variety of soil textures but the clayey type soils can be relatively harder to plough and pulverize sufficiently for the roots of young poppy plants to penetrate. On the other hand, sandy soils tend to lose water to percolation, resulting in the moisture being insufficient for the healthy growth.
Poppy prefers moderately cool weather and open sunny location; severely cold spell, frost, dull cloudy weather, high winds and very heavy rainfall during the lancing period adversely affect the quantity and quality of opium yield [17]. Water stress can affect the alkaloid production during various developmental stages of poppy plants, with sufficient supply water being beneficial for alkaloid accumulation in the capsules, while drought can increase the level of certain alkaloids. Nitrogen fertilization can elevate alkaloid accumulation [18] only under excessive light conditions, while severe drought can reduce the accumulation of morphinans [19].
Under pressure of the Nixon administration’s “war on drugs”, a crackdown on poppy cultivation was undertaken by the Thai government, deeming it illegal to grow the crop, with a reported drop in production from well over 10000 ha in 1961 to under 300 ha in 2015 [20]. Subsequent research and development of geographically suitable alternative crops and other incentives meant that the small landholders were dissuaded to cultivate the crop. This illegal status of the poppy crop meant that no substantial researches could be conducted about the plant and its medical benefits in Thailand. Additionally, there is almost no data about the morphological traits of poppy in Thailand and the herbarium specimens already outdated. With proven benefits of poppy already reported, the Office of the Narcotics Control Board, Thailand plans to study poppy for further usage in the pharmaceutical industry. In light of that effort, this study is a first comprehensive study of its kind reporting on the morphological characteristics of poppy growing naturally in Thailand. Thus, this study will add to the fundamental knowledge about the species in Thailand for future uses in the medical industry.
The manuscript objectives still need improvement and explanation in detail.
Please see Introduction.
The authors are advised to improve the manuscript in terms of adequate language levels as well as research paper structure.
See information and table below.
The authors should elaborate more on their findings and discussion compared to other studies, to their importance, as well as to plant profile in the screened region.
Please see conclusions.
English should be improved; grammar needs enhancement in many sentences and paragraphs.
Dr Tushar Andriyas has checked English language for this paper.
Figure 1, 5, and 10 are needed for resolution enhancement.
Figure 1, 5, and 10 are high resolution.
The discussion should show how these new findings support the research question and hypothesis.
Please see conclusions.
Opium poppy is illegal in most parts of the world including Thailand, such information about opium poppy is mostly lacking and its characteristics poorly known. We have got permission to study and support by the Narcotics Crop Survey and Control Institute, ONCB, Ministry of Justice. In this research project, we provide the morphology, taxonomy, anatomy, and palynology of Papaver somniferum. As a result, we update detailed morphological description for this species, which is more than information of previous studies in Thailand and neighboring countries. With proven benefits of opium poppy already reported, the Office of the Narcotics Control Board, Thailand plans to study opium poppy for further usage in the pharmaceutical industry.
The species can be characterised as a glabrous and glaucous herb; having unlobed or pinnately lobed leaves, and an amplexicaul base; variations in color and morphological characteristics of petals; and white filaments, occasionally purple with a white basal part, broadened at the apical part.
From our study, we found number of seeds per capsule of Papaver somniferum has a range of 593–4685. We realized number of seeds per capsule depends on size of fruits and number of carpels.
Please check the References in-text and end-list for uniformity in style.
We follow guideline for authors.

Reviewer 2 Report
Dear authors,
Congratulations on the search. However, I reiterate that the text presented in the manuscript needs to be more objective, in the same way that its tables and figures need to be reduced.

Author Response
Respond to Reviewer 2
The authors are grateful to thank the reviewer 2 for valuable comments.
Thank you very much for your kind consideration.
For improvements and corrections in this paper, the authors highlighted in red.
We add four paragraphs and nine references [12–20] in Introduction (Line 49–87).
Poppy is reported to have originated in ancient Mesopotamia (modern day Iraq and Kuwait) [12] and sourced for latex (milky sap) whose alkaloids are used in the pharmaceutical industry or for poppy seeds that are used in the food industry [13]. Poppy is illegal in most parts of the world including Thailand. Only a few countries that the legal cultivation of opium poppy for pharmaceutical and food industry is done in Australia, Canada, India, Central and Southern America, Türkiye (Turkey), Russia, Czech Republic, Slovakia, Holland, France, Hungary, Iran, Poland, Romania, and Spain [14,15].
Different cultivars with low alkaloid content can be used in the food industry for seed or oil production [16] such as in some European countries (e.g. Germany and the Czech Republic), which only permit the cultivation of ‘low‐morphine’ varieties of Papaver somniferum. Apart from these, some countries only grow varieties to derive pharmaceutical use from the capsules (e.g. Australia, France, and Spain) or for both culinary and pharmaceutical uses (e.g. Hungary and Slovakia). Poppy grows on a wide variety of soil textures but the clayey type soils can be relatively harder to plough and pulverize sufficiently for the roots of young poppy plants to penetrate. On the other hand, sandy soils tend to lose water to percolation, resulting in the moisture being insufficient for the healthy growth.
Poppy prefers moderately cool weather and open sunny location; severely cold spell, frost, dull cloudy weather, high winds and very heavy rainfall during the lancing period adversely affect the quantity and quality of opium yield [17]. Water stress can affect the alkaloid production during various developmental stages of poppy plants, with sufficient supply water being beneficial for alkaloid accumulation in the capsules, while drought can increase the level of certain alkaloids. Nitrogen fertilization can elevate alkaloid accumulation [18] only under excessive light conditions, while severe drought can reduce the accumulation of morphinans [19].
Under pressure of the Nixon administration’s “war on drugs”, a crackdown on poppy cultivation was undertaken by the Thai government, deeming it illegal to grow the crop, with a reported drop in production from well over 10000 ha in 1961 to under 300 ha in 2015 [20]. Subsequent research and development of geographically suitable alternative crops and other incentives meant that the small landholders were dissuaded to cultivate the crop. This illegal status of the poppy crop meant that no substantial researches could be conducted about the plant and its medical benefits in Thailand. Additionally, there is almost no data about the morphological traits of poppy in Thailand and the herbarium specimens already outdated. With proven benefits of poppy already reported, the Office of the Narcotics Control Board, Thailand plans to study poppy for further usage in the pharmaceutical industry. In light of that effort, this study is a first comprehensive study of its kind reporting on the morphological characteristics of poppy growing naturally in Thailand. Thus, this study will add to the fundamental knowledge about the species in Thailand for future uses in the medical industry.
Respond to Reviewer 2
Line |
Original |
Corrections |
20 |
the International Union for Conservation of Nature (IUCN) conservation assessment |
We deleted this part |
34–35 |
Keywords: I suggest deleting both, as they are in the title. |
We deleted the word “Anatomy” and “taxonomy” in the keywords. |
43–49 |
The genus…. |
We confirm that these characteristics suitable for the genus. |
50–89 |
Introduction page 2 |
We add four paragraphs and nine references [12–20]. |
109 |
Please reference the figures here. |
2.1.1. Papaver somniferum L., Sp. Pl. 1: 508. 1753. (Figures 1–9). |
119 |
Delete Habit |
We deleted “Habit”. |
121 |
Unnecessary. and fruit stalks (fruits having distinct latex color). |
We deleted “and fruit stalks (fruits having distinct latex color)”. |
123 |
Delete terete; It is the same as cylindrical. |
We deleted “terete”. |
125 |
Unnecessary. All the leaves come from the stem. |
We would like to retain the word “cauline”, it is significant character for this grass species. |
125 |
Delete (pinnatifid); It is the same as pinnately lobed. |
We deleted “(pinnatifid)”. |
128 |
both surfaces glabrous and glaucous |
glabrous and glaucous on both surfaces |
133 |
Delete (a single flower); It is the same as solitary. |
We deleted “(a single flower)”. |
133 |
Delete (radially symmetrical); It is the same as actinomorphic. |
We deleted “(radially symmetrical)”. |
138–139 |
Delete (sepals falling off before being fully opened flowers) |
We would like to retain (sepals falling off before being fully opened flowers). Caducous is different from “sepals falling off before being fully opened flowers.” |
144 |
- |
We add “Figures 6–9”. |
200 |
Figure 1 (A) Delete stem, leaves, flower bud, opened flower, and fruit |
We add “fertile branch”. |
200 |
Figure 1 (B) Delete opened |
We deleted “opened”. |
205–206 |
Figure 1 (p. 4) |
We add Materials from C. Ngernsaengsaruay et al. Ps03-06012023 (BKF) and C. Ngernsaengsaruay et al. Ps04-06012023 (BKF). |
255 |
Figure 2 (A–B) Delete stems, leaves and flowers |
We deleted “stems, leaves and flowers”. |
255 |
Figure 2 (C) underground system ? |
It is taproot. |
255–256 |
Figure 2 (D–G) |
(D–G) leaf shapes: lanceolate, lanceolate-ovate, ovate, and broadly ovate |
284 |
- |
We add “593–4685 seeds per capsule,” |
288 |
Delete Variability in the color and morphological characteristics of the flowers of Papaver somniferum is shown in Table 1, |
We deleted “Variability in the color and morphological characteristics of the flowers of Papaver somniferum is shown in Table 1,” |
290 |
Reduce this part significantly, leaving only what is essential for species recognition. As it is, it repeats the description of the species for the most part. |
We deleted “all paragraph of Recognition.” |
291 |
|
We deleted “(stiff hairs)”. |
292 |
|
We deleted “pinnately lobed”. |
292–293 |
|
We deleted “(deeply pinnately lobed)”. |
300–311 |
Distribution |
We cited the literature. Should be not shorten. |
|
Figure 2 (p. 5), Figure 3 (p.7), Figure 4 (p. 8) |
We would like to retain Figure 2, 3, and 4 showing vegetative and reproductive parts. |
|
Figure 5 (p. 9) |
To show number of stigma lobes per flower, it is easy to read. |
|
Table 1 (p. 9) |
We would like to retain Table 1 because the description no detail about variability in the flower color and also easy to read. Cannot deleted. |
|
Figure 6 (p. 10), 7 (p. 11), 8 (p. 12), 9 (p. 13) |
We would like to retain Figure 6, 7, 8, and 9 to show variability in the color of the petals (flowers), outer pair petal, inner pair petal, and morphological characteristics |
|
Table 2 (p. 14) |
We would like to retain Table 2 showing sample size, range and mean ± SD, it is easy to read. |
783–784 |
Pollinator |
We add reference “using online database [31]”, |
927 |
Stem anatomy |
We add “are circular outline and” in the first paragraph. We have already mentioned about below epidermis; number of parenchyma layer; vascular bundle type; and pith (We highlighted in blue). |
936 |
Leaf lamina anatomy |
We have already mentioned as your comments. |
946 |
Transverse section of leaf lamina |
We deleted “bifacial, and is protected by the cuticle”. |
|
Figure 12 (p. 20) |
We changed photos. |
1034 |
Figure 12 (D) adaxial epidermis |
epidermal cells on the adaxial surface showing polygonal shape |
1034–1035 |
Figure 12 (E–I) abaxial epidermis |
epidermal cells on the abaxial surface showing polygonal and irregular shapes |
|
Figure 13 (p. 21) |
We changed photos. |
1086, 1089 |
Figure 13 (C–D) |
(C–E) |
1087–1088 |
Figure 13 |
Materials from C. Ngernsaengsaruay et al. Ps03-06012023 (BKF) and C. Ngernsaengsaruay et al. Ps04-06012023 (BKF) |
1122–1126 |
Discussion. Shorten this paragraph. |
We deleted “sometimes oblate spheroidal” in this paragraph. |
1127–1134 |
Discussion. |
We deleted “P diameter with a mean of 31.14 ± 3.35 µm; E diameter with a mean of 30.20 ± 3.33 µm; amb diameter with a mean of 30.65 ± 3.35 µm; colpus length with a mean of 21.77 ± 3.96 µm; colpus width with a mean of 11.06 ± 1.86 µm; and exine thickness with a mean of 2.02 ± 0.39 µm.” in this paragraph. |
1123 |
Acknowledgments |
We deleted “This research was funded by Office of the Narcotics Control Board, Ministry of Justice.” |
1173 |
Conclusions |
We deleted four paragraphs of conclusions. We rewrite three paragraphs of conclusions |

Round 2
Reviewer 2 Report
The authors responded to most of my suggestions for adjustments and/or modifications. Therefore, I agree with the publication of the manuscript.
Yours sincerely